# Enhanced Anti-Skin-Aging Activity of Yeast Extract-Treated Resveratrol Rice DJ526

**DOI:** 10.3390/molecules27061951

**Published:** 2022-03-17

**Authors:** Vipada Kantayos, Jin-Suk Kim, So-Hyeon Baek

**Affiliations:** Department of Agricultural Life Science, Sunchon National University, Suncheon 59722, Korea; vkwah@naver.com (V.K.); kimjs6911@naver.com (J.-S.K.)

**Keywords:** elicitor, resveratrol, antioxidant, melanogenesis, wound healing, anti-inflammation

## Abstract

Resveratrol is a powerful antioxidant that defends against oxidative stress in cells but is not found in large quantities in plants. Resveratrol-enriched rice DJ526, which was developed as a functional crop, shows a diverse range of biological activities. Resveratrol production is measured as total resveratrol and its glycoside, piceid, which is mainly found in plant-derived resveratrol. In the present study, elicitation using yeast extract (YE), methyl jasmonate, and jasmonic acid increased resveratrol production in DJ526 rice seeds. DJ526 seeds elicited using 1 g/L (YE1) and 5 g/L yeast extract (YE5) showed enhanced resveratrol production and antioxidant activity. YE5-treated DJ526 seeds showed decreased melanin content by 46.1% and 37.0% compared with the negative control and DJ526 (non-elicitation), respectively. Both YE1 and YE5 efficiently improved the wound-healing activity by reducing the wound gap faster than in untreated cells, with a maximum rate of 60.2% at 24 h and complete closure at 48 h. YE1 and YE5 significantly decreased the levels of proinflammatory cytokine, TNF-α, and enhanced collagen synthesis in inflammatory cells. These findings indicate that YE-treated resveratrol rice DJ526 may improve resveratrol production and could be an active antiaging ingredient for cosmetic and skin therapy applications.

## 1. Introduction

Resveratrol is an antioxidant compound found in red wine, which is its most recognized source [1]; however, it is not found in large quantities in plants. Resveratrol and its glycosides have a wide range of health benefits, from cardioprotection to antiaging [2]. Recently, resveratrol-enriched rice DJ526 was developed via genetic modification, with the objective of developing a crop-derived resveratrol product with medicinal properties [3]. Potential therapeutic uses of DJ526 rice have been described, such as anti-obesity, improving lifespan, skin whitening, and inhibiting neuroinflammation [4,5,6,7,8]. DJ526 seeds contain around 2 μg/g of resveratrol in the callus [9] and 1.4–1.9 µg/g in the seed grain [10], and the concentration of resveratrol has been reported to sequentially increase during germination [11].

The emergence of plant biotechnology as a modern tool has created a connection between plants and humans. Bioactive compounds derived from plants are, therefore, essential to human wellness, such as nutraceuticals, pharmaceutical products, and cosmetic ingredients, as these compounds are not harmful and have low side effects compared with other synthetic chemicals [12]. Elicitation is an approach that has been used to enrich the bioactive compound content in plants. It is a process that uses a stimulant (elicitor) to stimulate plant responses by increasing the compound yield. Resveratrol can be triggered by elicitors, such as jasmonic acid (JA) and chitosan, as well as ultrasound treatment, and has mostly been studied in cell suspension culture [13,14]. However, increasing the resveratrol yield in DJ526 seeds through elicitation is an effective tool and creates a good opportunity for DJ526 rice to be used as a new material source in plant-based pharmaceuticals.

Skin is a complex organ that consists of many types of cells and tissues that serve to defend against pathogens and ultraviolet light and maintain fluid balance in the body [15]. Hyperpigmentation is a skin condition derived from the overproduction of melanin, which is produced by melanocytes (dendritic cells) via melanogenesis in the skin layers. Exposure to UV radiation increases melanin production, which may be a disturbing problem in daily life. The appearance of mottled skin due to photo exposure contributes to premature skin aging. Therefore, it is important to thoroughly understand the mechanism of melanin production [16]. Melanogenesis is regulated by various enzymes, including tyrosinase, TRP-1, TRP-2, and microphthalmia-associated transcription factor (MITF) [17]. There are several types of hyperpigmentation, including melasma, which is the most common pigmentation disorder that causes dark patches on the skin. The production of melanin is associated with inflammation and is commonly seen in skin prone to inflammation [18]. There is interest in developing drugs and cosmetics products that reduce melanin production as well as its anti-inflammatory effects. Also, extensive research in wound healing may show cosmetic effects and increase the impact of the skin-therapeutic opportunities of DJ526 rice.

DJ526 rice has shown diverse ranges of pharmaceutical benefits; therefore, the present study aimed to examine the effects of elicitors on the production of resveratrol, including the glycoside, piceid, in germinated DJ526 seeds. The study evaluated the antioxidant activity and anti-skin-aging activity, including anti-melanogenesis, wound healing, and anti-inflammatory effects in skin cells (melanocytes and keratinocytes). We aimed to identify which elicitors improved resveratrol production and showed increased skin-therapeutic opportunities in DJ526 seeds.

## 2. Results

### 2.1. Effect of Elicitors on Total Resveratrol Production and Antioxidant Activity of Elicitor-Treated DJ526 Seeds

In the present study, total resveratrol production was defined as the amount of resveratrol and its glycoside, piceid. The effect of elicitor-treated DJ526 seeds on resveratrol production was examined by HPLC compared with a standard mixture of piceid and resveratrol. DJ526 seeds (non-elicitation) contain around 56.8 µg/g of total resveratrol product. Among the elicitor-treated DJ526 seeds, YE1 and YE5 treatment led to high amounts of total resveratrol production. Treatment with YE5 showed the highest amount of total resveratrol production, with up to 50.7% (85.6 μg/g; 25 μg/g resveratrol and 60.6 μg/g piceid), whereas YE1 produced 19.2% (67.7 μg/g; 19.72 μg/g resveratrol and 48.0 μg/g piceid) compared with control DJ526 seeds (Figure 1a). Chromatograms are shown in Appendix A. The total resveratrol production in response to YE treatment was confirmed by the gene expression of RS, which was upregulated by 20.7% following treatment with YE1 and 80% with YE5 treatment (Figure 1b). There were no significant differences in the concentration of resveratrol following treatment with 0.01 mM and 0.1 mM MJ and JA or control (DJ526 seeds without elicitation).

The antioxidant activity of elicitor-treated DJ526 seeds extracts ranged from 3.9% to 94.3% (Table 1). YE5 exhibited the highest ABTS radical scavenging activity followed by YE1 > DJ526 > MJ0.1 > JA0.1 > JA0.01 > MJ0.01 in a concentration-dependent manner (Table 1). The percentage antioxidant activity corresponding to 10–100 mg/L YE5, MJ0.1, JA0.01, and JA0.1 showed no differences in antioxidant activity. Among these, YE1 and YE5 exhibited greater radical scavenging activity against ABTS radicals (antioxidant activity) compared with other treatments. Furthermore, 10 mg/mL YE5 possessed a scavenging activity similar to that of 100 mg/mL YE5, whereas other treatments showed no differences. The antioxidant capacity was also measured in terms of the ascorbic acid (vitamin C) equivalent antioxidant capacity (VEAC) by comparing the calibration curve of vitamin C (an essential antioxidant) obtained from plotting the percentage of antioxidant activity compared with the concentration (3.9065 μg/mL–0.2500 mg/mL, R^2^ = 0.9905) (Appendix A). Percentage antioxidant activity and VEAC were positively correlated (Pearson’s correlation coefficient = 0.531, *p* < 0.01).

The concentration that can inhibit ABTS radical 50% (IC_50_) was calculated and is shown in Table 1, where a lower IC_50_ indicates a highly potent antioxidant. Based on the result, the IC_50_ value also correlated with the percentage of antioxidant activity as well as the VEAC value.

### 2.2. Effect of YE-Treated DJ526 Seeds on the Antimelanogenesis Activity of Malan-a Cells

YE treatment significantly increased resveratrol production in terms of the yield, the transcription level of the RS gene, and the ABTS radical scavenging activity. Therefore, YE1 and YE5-treated DJ526 seeds were selected for in vitro cell-based experiments. Anti-melanogenesis was examined in melan-a producing-cells. Cell viability was measured after the cells were exposed to crude extracts for 72 h to test the cytotoxicity of the extracts on cells at various concentrations. Cells were divided into two groups: non-elicitation treatment (DJ526) and elicitation with yeast extracts (YE1 and YE5) at different concentrations. The vehicle control group (DMSO) and positive control (arbutin, a skin depigmenting agent) were monitored along with the experimental groups. None of the treatments were cytotoxic and 1–100 mg/L of tested extracts were considered safe for melan-a cells (Figure 2a).

The skin-whitening effect of YE was evaluated by measuring the melanin content and tyrosinase activity in melan-a cells. Figure 2b,c shows the effect of 1–100 mg/mL of YE1 and YE5 on the melanin content. YE treatment significantly reduced the melanin content (as a percentage of the control) in a dose-dependent manner, and its efficiency was comparable to that of arbutin (positive control) at the same concentrations (10 and 100 mg/mL). On the other hand, low concentrations (1 mg/mL) did not reduce the melanin content. Concentrations of 10 and 100 mg/mL YE1 decreased the melanin content by 32.2% and 26.7% of the control (DMSO), which was similar to arbutin, and 23.1% and 16.6% compared with DJ526 at the same concentrations, respectively. Concentrations of 10 and 100 mg/mL YE5 showed strong anti-melanogenic activity by decreasing the melanin content to 46.1% and 43.4% that of the control (DMSO) and 37.0% and 33.3% compared with DJ526 at the same concentrations, respectively.

Tyrosinase activity in DJ526 and YE-treated DJ526 is shown in Figure 2d. The results show that treatment with 10 mg/mL of YE1 and YE5 inhibited tyrosinase activity by 53.4% and 75.9%, respectively. YE5 treatment showed a stronger tyrosinase inhibitory activity than arbutin, in which inhibited tyrosinase activity by 64.3% of the control. On the other hand, the tyrosinase activity of DJ526 extract showed no differences at any of the concentrations. These results indicated that 10 mg/mL of YE treatment of DJ526 seeds was the lowest dose that strongly reduced the melanin content and tyrosinase activity. Since treatment at this concentration was more effective and showed moderate-to-good inhibitory effects on melanin production and tyrosinase activity at the cellular level, it was selected for further studies to examine melanogenesis activity at the transcriptional and protein expression levels.

mRNA expression of melanogenesis-related genes, including MITF, tyrosinase, TRP-1, and TRP-2, was evaluated after 72 h of treatment with 10 mg/mL of DJ526, YE1, and YE5. DJ526 and YE treatments suppressed MITF, which is a transcription factor that regulates the activation of tyrosinase during melanogenesis (Figure 3a). DJ526, YE1, and YE5 treatment downregulated tyrosinase expression by 9.1%, 12.6%, and 19%, respectively, compared with the control. In addition, YE5 markedly downregulated TRP-1 by 35.4% of the control, while downregulation of TRP-2 expression was observed after DJ526 treatment. The expression of MITF, tyrosinase, TRP-1, and TRP-2 proteins were examined using Western blot analysis and protein expression was normalized using GAPDH as an internal control (Figure 3b). MITF was suppressed 0.63–0.69-fold in response to DJ526, YE1, and YE5 treatment. Tyrosinase was significantly downregulated (0.6–0.24-fold) by YE5. Only YE5 significantly downregulated TRP-1 (0.75-fold, *p* < 0.05), and TRP-2 was not affected by any of the treatments.

### 2.3. In Vitro Wound-Healing Effects of YE-Treated DJ526 Seed Extract on Human HaCaT Keratinocytes

The HaCaT human keratinocyte cell line was used as a cell-based in vitro wound healing model to examine the cell viability of cells exposed to DJ526, YE1, YE5, and vitamin C (positive control) using an Ez-Cytox protocol previously described [8]. Most treatments (DJ526, YE1, YE5, and vitamin C) at concentrations of 1, 10, and 100 mg/mL had no cytotoxic effects on HaCaT cells compared with the untreated control (Figure 4a), except 100 mg/mL vitamin C, which showed cytotoxic effects on HaCaT cells (data not shown). Figure 4a illustrates the wound closure effects of DJ526, YE1, and YE5 treatments on HaCaT cells at 0, 24, and 48 h compared with the untreated control and vitamin C. In the first 24 h period, 10 mg/mL YE1 and YE5 treatments rapidly decreased the wound gap compared with other treatments. The highest rate of wound closure was 60.2%, which was seen with the 10 mg/mL YE5 treatment, followed by 10 mg/mL of YE1 (56.1%). At 48 h, the wounds almost closed completely, with a maximum rate of 78.2–100%, except in the untreated cells. Wound closure was achieved in response to all experimental treatments (except untreated cells) by inducing cell migration.

### 2.4. YE-Treated DJ526 Extracts Downregulate TNF-α in TNF-α-Induced HaCaT Cells

YE1 and YE5-treated DJ526 extracts showed great potential to promote cell proliferation and wound healing in HaCaT cells. The effectiveness of reducing inflammation of YE treatments was determined by downregulating the proinflammatory cytokine, TNF-α, and inflammatory mediators (matrix metalloproteinase; MMP)-related genes in TNF-α-induced HaCaT cells (Figure 5). Treatment with 100 mg/mL of YE1 and YE5 decreased TNF-α expression around 4.3-fold (*p* < 0.05) and 2.6-fold, respectively (*p* < 0.05) compared with the control (TNF-α-treated HaCaT cells). Furthermore, YE treatments showed suppressive effects on MMP-9 (a key inflammatory mediator) in TNF-α-induced cells in a dose-dependent manner. MMP-2 was downregulated in TNF-α-treated cells and 10 mg/mL of YE5 showed a similar expression relative to cells not exposed to TNF-α. The collagen degradation effector (MMP-13) was downregulated in cells treated with TNF-α and exposed to YE1 and YE5. Moreover, the gene expression of COL1A was upregulated around 2.6-fold in YE5 and 0.4-fold in YE1-treated cells compared with control and TNF-α treated cells. Improvements in skin barrier properties are represented by an upregulation of filaggrin. In TNF-α treated cells, filaggrin was downregulated 2.5-fold compared to that of control cells. However, among the TNF-α-treated cells, YE5 treatment showed a 1.3-fold increase in expression levels over TNF-α treated cells (inflammatory cells), although the expression of filaggrin was lower than in control cells (Figure 5).

## 3. Discussion

The present study examined the effects of elicitors of resveratrol production in five-day-old DJ526 seeds. This stage has previously been shown to be associated with high resveratrol production [11]. We showed that extracts from this germination stage (5 days) potentially promote biological activities. DJ526 seeds contain *trans*-resveratrol and are usually associated with a higher amount of *trans*-piceid in seed grain due to glycosylation. This process is an important reaction that determines the biological activity of resveratrol. During glycosylation, resveratrol can be converted to various glycosylate forms, such as pterostilbene, *trans*-piceatannol, piceatannol, astringin, and piceid. Piceid is the most prevalent form found in plant-containing resveratrol [19], indicating that the conversion of resveratrol to piceid is a crucial process to enhance its bioavailability and biological activity. Moreover, resveratrol and piceid have been shown to express a wide range of biological activities, such as antioxidant activities, antitumor, and anti-inflammatory effects, in both in vitro and in vivo studies in animals and humans [20,21,22]. The bioavailability assessment of the combination of these compounds can be more effective than the study of a single compound. Therefore, the present study examined resveratrol production in terms of total *trans*-resveratrol and *trans*-piceid content.

The potential therapeutic application of DJ526 rice has been well reported; however, the use of elicitation to induce resveratrol production and its bioactivity for cosmetic purposes is not well understood. Therefore, we used several elicitors to improve resveratrol production in DJ526 seeds by elicitation. Elicitation is a tool for increasing the content of phytochemical compounds in many plants. It is used to promote higher levels of active substances in plants than under normal conditions using plant defense mechanisms [23]. Elicitation with 1 and 5 g/L of YE led to greater resveratrol production than non-elicitation (DJ526) and other elicitors, as confirmed by HPLC analysis and the increased gene expression of RS. The highest resveratrol yield after treatment with YE5 in 5-day-old germinated seed was 25 μg/g, which is similar to the amount of resveratrol in grape skin (24 μg/g), while germinated peanut contained 0.638 μg/g at day 8 [24,25]. Thus, the resveratrol content varies among plants, plant parts, and stages of growth as well as with glycosylation, which can convert resveratrol to other derivative forms.

The present study showed a good capacity of DJ526 seeds and elicitor-treated DJ526 seed extracts as natural antioxidants. Crude extracts of DJ526 and elicitor-treated DJ526 seeds have significant antioxidant effects as can be seen by the percentage of ABTS-scavenging activity and VEAC values. The in vitro antioxidant activity of elicitor-treated DJ526 seed extracts indicated that YE-treated DJ526 seeds exhibited the most potent antioxidant activity among the elicitors and non-elicitation treatment. The antioxidant capacity of DJ526 and elicitor-treated DJ526 seed extracts was related to the presence of resveratrol product. Nonetheless, antioxidant capacity is dependent on several factors, such as the reaction rate of the sample, the type of reactive species, and the concentration ratio between the antioxidant and target [26]. The antioxidant capacity of both resveratrol and piceid depends on the type of reactive oxygen. Piceid was reported to have a stronger antioxidant activity than resveratrol, as measured by DPPH assay, but resveratrol showed a stronger ABTS-scavenging activity than piceid [27]. Similarly, the ABTS-scavenging activity of isolated resveratrol was higher than mulberry-derived piceid; however, both resveratrol and piceid are powerful antioxidants [28].

The production of melanin relies on three major enzymes: tyrosinase, TRP-1, and TRP-2, which are regulated by the transcription factor MITF in the nucleus. The biosynthesis of melanin is initiated in cytoplasm by the reaction of enzyme tyrosinase and oxidizes L-tyrosine to L-dopa and dopaquinone. Then, the process of melanin synthesis takes place in the melanosomes. Dopaquinone works as a precursor of intermediate molecules in the melanogenesis pathway in melanosomes, including indole-2-carboxylic acid-5,6-quinone (ICAQ), 5,6-dihydroxyindole-2-carboxylic acid (DHICA), and indolequinone (IQ), which are further polymerized to eumelanin. Apart from tyrosinase, TRP-1 and TRP-2 are critical enzymes in the melanin formation process in melanosomes. However, the reduction of melanin production was established by the inhibition of tyrosinase activity, which is a rate-limiting enzyme involved in melanogenesis [29]. Although, melanin production is required for several responsive elements, such as TRP-1, TRP-2, and the transcription factor MITF, reducing tyrosinase activity is key to the development of anti-pigmentation agents [30]. Our findings showed that YE5 treatment of DJ526 seeds had anti-melanogenesis effects by reducing tyrosinase activity at the gene and protein expression levels in melan-a cells. However, MJ treatment at a concentration of 5 µM demonstrated an excellent whitening effect in the DJ526 callus [8].

Both YE1 and YE5 treatments promoted wound healing by inducing the cell migration of human keratinocytes (HaCaT cells). The critical period for the wound closure rate was the first 24 h, in which DJ526 and YE treatments showed a rate higher of wound closure than the untreated and the vitamin C (positive control) groups. The keratinocytes almost completely closed the wound gap after 48 h following DJ526 and YE treatments in a concentration-dependent manner. Wound healing generally occurs after tissue injury and consists of four stages of transformation, namely homeostasis, inflammation, proliferation, and maturation or remodeling, which involve multiple regulators including proinflammatory cytokines, such as TNF-α. TNF-α is an important player in inflammation and skin barrier dysfunction [31,32,33]. Among the TNF-α-induced cells, 10 mg/mL of YE5 and DJ526-treated cells maintained barrier function better that other TNF-α treatments. The anti-inflammatory activity of resveratrol and its derivatives, such as piceid and oxyresveratrol, provides protection to cells by reducing the expression of inflammatory cytokines and its mediators (MMPs) in inflammatory-induced cells [34,35,36]. The use of resveratrol in cosmetic formulas can increase collagen production in dermal cells [37]. Our study also revealed that YE-treated DJ526 seeds that produce high amounts of resveratrol can potentially promote collagen production in human keratinocytes. Resveratrol and piceid (or polydatin) can promote anti-skin-aging by activating sirtuin, which is a crucial factor that delays cell aging [38,39]

## 4. Materials and Methods

### 4.1. Preparation of Elicitor-Treated DJ526 Seed Extract

Five-day-old DJ526 seeds were selected for the study as they have been shown to contain high quantities of resveratrol [10]. The seeds were induced to germinate on 2N6 media containing elicitor treatments, including 1 and 5 g/L of yeast extract (YE1 and YE5, respectively), methyl jasmonate (MJ; 0.1 and 0.01 mM), and JA (0.1 and 0.01 mM). Elicitor-treated DJ526 seeds were harvested and rinsed with water to remove excess culture media and the seeds were dried using a hot-air oven method at 60 °C for 48 h and ground into a fine powder using an electric mixer. The sample powder was extracted using 80% methanol under sonication for 1 h. The solvent was removed using a rotary evaporator at 50 °C followed by lyophilization to concentrate the crude extracts. Crude extracts were prepared at concentrations of 1, 10, and 100 mg/mL before being used for animal cell experiments.

### 4.2. Effect of Elicitation on Resveratrol and Piceid Content

High performance liquid chromatography (HPLC) analysis of elicitor-treated DJ526 seed extracts was performed using a Waters e2695 separation module (Waters Pacific Ptd. Ltd., Singapore) equipped with a C18 column (4.6 × 150 mm; Waters, Milford, MA, USA). Crude extracts were diluted in 80% methanol and filtered through a 0.45 μm sterile syringe filter. Data were analyzed using the Empower chromatography software. Chromatographic separation was performed under a gradient program using distilled water (solvent A) and acetonitrile (solvent B) as a mobile phase solvent. The gradient program was as follows: 0–37 min, 10% solvent B; 37–38 min, 30% solvent B; 38–45 min, 100% solvent B; 45–50 min, 10% solvent B. Flow rate was set at 1 mL/min, with an injection volume of 10 μL. Elutes were monitored at 308 nm, with a total analysis time of 40 min.

### 4.3. Measurement of Antioxidant Activity of Elicitor-Treated DJ526 Seed Extract by 2,2′-Azino-bis (3-ethylbenzothiazoline-6-sulfonic acid) Diammonium Salt (ABTS) Scavenging Assay

The antioxidant activity of elicitor-treated DJ526 seed extracts was based on the radical scavenging capacity of extracts that decolorized the radical solution. Standard ABTS radical (ABTS^●+^) reagent was freshly prepared, as described previously, with some modifications [40]. ABTS^●+^ reagent was obtained by the activation of ABTS with potassium persulfate. Next, 7 mM ABTS (0.0384 g) and 2.45 mM potassium persulfate (0.0066 g) were dissolved in 10 mL of distilled water. A 1:1 (*v*/*v*) solution of ABTS and potassium persulfate was mixed and incubated in the dark for 12–16 h. A working ABTS^●+^ solution was prepared by diluting the 1:1 ABTS and potassium persulfate solution until the absorbance at 734 nm was 0.70 ± 0.02. Elicitor-treated DJ526 seed extracts were prepared at concentrations of 1–100 mg/L. Next, 20 mL of extract was mixed with 180 μL of ABTS^●+^ reagent and plated on 96-well plates. Reaction mixtures were incubated in the dark at room temperature for 7 min before measuring the absorbance. ABTS^●+^ reagent without extract was used as a blank. The percentage scavenging activity was calculated as follows:% scavenging activity = (A_734_ blank − A_734_ treatment)/A_734_ blank × 100(1)
where A_734_ represents the absorbance at 734 nm.

The effective concentration of the extract that scavenges 50% of the ABTS radical [18] or IC_50_ was also calculated to compare antioxidant activity in each treatment.

### 4.4. In Vitro Cell Culture and Treatment

Melan-a cells were obtained from Prof. Sun-Yeou Kim (College of Pharmacy, Gachon University, #191, Hambakmoero, Incheon, Korea). Cells were cultured in RPMI 1640 media supplemented with 10% fetal bovine serum (FBS), 10% penicillin–streptomycin, and 200 nM of 12-*O*-tetradecanoylphorbol-13-acetate. HaCaT cells were cultured in Dulbecco’s modified Eagle medium (DMEM) supplemented with 10% FBS and 10% penicillin–streptomycin. All cells were maintained in cell culture conditions of 37 °C, 5% CO_2_. Sample extracts (stock solution 1–100 mg/mL) were used for cell-based in vitro experiments, including the anti-melanogenic activity of melan-a cells and the anti-inflammatory effect of tumor necrosis factor (TNF-α)-induced HaCaT cells, and the wound-healing assay in HaCaT cells. Stock solutions were diluted in culture medium with 0.1% DMSO (vehicle agent).

### 4.5. Cell Viability Assay

Cell viability was measured using a water-soluble formazan through a succinate–tetrazolium reductase system with an Ez-Cytox kit (DoGen Bio, Seoul, Korea). Briefly, cells were seeded into 96 wells at a density of 2 × 10^4^ cells/mL with a final volume of 100 µL per well, cultured for 24 h, and then incubated with crude extract for 72 h. The effect of TNF-α and its combination with crude extracts on the cytotoxicity of HaCaT cells was measured in DMEM serum-free media. Cell viability was measured by adding 10 µL of Ez-Cytox solution to each well and incubating for 4 h before measuring the absorbance at 450 nm. The percentage cell viability was calculated as follows:% cell viability = (A_405_ treatment − A_405_ blank)/(A_405_ DMSO − A_405_ blank) × 100(2)
where A_405_ represents the absorbance at 405 nm.

### 4.6. Measurement of Melanin Content and Cellular Tyrosinase Activity

Melan-a cells were seeded into 6-well plates at a density of 8 × 10^4^ cells/mL and cultured for 24 h before treatment with extracts. Cells were treated with crude extracts (1, 10, and 100 mg/mL) and incubated under culture conditions (at 37 °C, 5% CO_2_). After 72 h, cells were harvested and washed twice with phosphate-buffered saline (PBS). Cells were detached from the culture dish using trypsin–EDTA (1×) and then pelleted by centrifugation at 2000 rpm at 4 °C for 10 min. Pellets were washed with cold PBS and lysed with lysis buffer (0.5 M EDTA, 1% Triton-X, and 0.1 M sodium phosphate with a pH of 6.8) for 30 min on ice and centrifuged at 13,000 rpm for 30 min. The melanin content was measured in separated pellets, whereas the supernatant was used to measure cellular tyrosinase activity. The supernatant was transferred into a new tube, then the pellets were washed with 100% ethanol and dissolved with 1N NaOH at 80 °C for 1 h. The absorbance was measured at 405 nm and the melanin content was calculated as a percentage of the control (DMSO). Tyrosinase activity was measured by measuring protein (40 µg) in the supernatant by Bradford assay and adjusting the volume with lysis buffer. L-DOPA (2 mg/mL) was added to the lysate and incubated at 37 °C for 1 h and the absorbance was measured at 475 nm.

### 4.7. Quantitative Polymerase Chain Reaction (qPCR) Analysis

Quantitative PCR was used to determine the gene expression of resveratrol synthase (RS), melanogenesis-related genes in melan-a cells, inflammatory cytokine production in TNF-α-induced HaCaT cells, and type 1 collagen in HaCaT cells. RNA extraction was performed using TRI reagent (Invitrogen, Carlsbad, CA, USA). A total of 1 μg of eluted RNA per sample was reverse transcribed to cDNA using a Power cDNA synthesis kit (Intron). Reaction mixture was prepared by mixing RealMODTM Green W2 2× qPCR mix (Intron, Korea), primers (Appendix A), cDNA template, and 18.2 MΩ·cm water and analyzed on a qPCR machine (CFX96 Real-time PCR detection system, BioRaD, CA, USA). mRNA expression was analyzed using CFX Maestro™ software. Actin, GAPDH, and HPRT1 were used as housekeeping genes for RS expression in DJ526 rice, melan-a cell (mouse melanocytes), and HaCaT cell (human keratinocytes) experiments, respectively.

### 4.8. Western Blot Analysis

Melan-a cell lysates were prepared using RIPA buffer supplemented with 1% phenylmethylsulfonyl fluoride and 1X protease inhibitor cocktail (Sigma-Aldrich, St. Louis, MO, USA). Lysates (40 µg) were resolved by SDS-PAGE and transferred to a nitrocellulose membrane. Proteins were measured using antibody-based probes, including anti-tyrosinase, anti-TRP-1, anti-TRP-2, and anti-MITF (Santa Cruz biotechnology, Dallas, TX, USA). GADPH was used as a loading control antibody.

### 4.9. Wound-Healing Scratch Assay

HaCaT cells were obtained from Prof. Sun-Yeou Kim (College of Pharmacy, Gachon University, #191, Hambakmoero, Incheon, Korea). Cells (4 × 10^5^ cell/mL) were used as a model for stimulatory cell migration effects. Cells were seeded in 24-well plates and pretreated with plant extracts for 24 h before introducing a wound by scratching [41]. Cells were grown to around 95% confluence then vertically scratched using a 200 µL sterile micropipette tip. The cell debris was removed by rinsing twice with serum-free culture media until the debris was not observed, then DJ526 and yeast extract-treated resveratrol rice extracts (YE1, YE5) were applied at concentrations of 1, 10, and 100 mg/mL per well. Cell migration and wound closure were observed at 0, 24, and 48 h. Vitamin C was used as a positive control for this experiment.

### 4.10. Statistic Analysis

Data are presented as the mean ± standard deviation (SD). Statistical differences were evaluated by analysis of variance (ANOVA) using the IMB SPSS statistics 26 software (IBM, Armonk, NY, USA). Comparisons between two groups were performed using Student’s *t*-test. All *p*-values < 0.05 were considered to be statistically significant.

## 5. Conclusions

The findings of the present study indicate that YE1 and YE5 induced higher amounts (67.7 and 85.6 μg/g, respectively) of resveratrol production (total of *trans*-resveratrol and *trans*-piceid) than other elicitors in DJ526 seeds (resveratrol-enriched rice). MMJ and JA had no effects on resveratrol production compared with non-elicited DJ526. YE-treated DJ526 seeds possessed a strong ABTS radical scavenging activity and excellent anti-skin-aging functions in an in vitro study of skin cells, including anti-melanogenesis, wound healing, and anti-inflammatory properties. Melanin production in melan-a cells after treatment with YE-treated DJ526 seed extract was reduced via downregulation of tyrosinase activity. The wound-healing activity of these extracts showed a higher wound closure rate than vitamin C (positive control) and non-elicited D526. Furthermore, YE-treated DJ526 seed extract acted as an anti-inflammatory agent in TNF-α-induced HaCaT cells (human keratinocytes) by suppressing inflammatory mediators, such as TNF-α, MMP-2, MMP-9, and MMP-13, and increased collagen synthesis in inflammatory cells. The results of the present study suggest that YE-treated DJ526 seeds may be an effective enhancer of a phyto-derived active compound with anti-skin-aging properties.

## Figures and Tables

**Figure 1 molecules-27-01951-f001:**
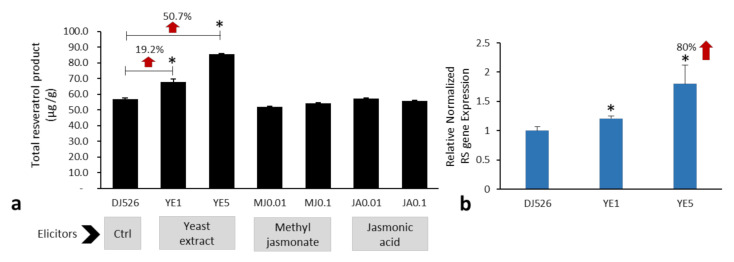
Resveratrol bioproduct and RS gene expression in elicitor-treated DJ526 seeds, including 1 and 5 g/L yeast extract (YE1, YE5), 0.01 and 0.1 mM methyl jasmonate (MJ0.01, MJ0.1), and 0.01 and 0.1 mM jasmonic acid (JA0.01, JA0.1). (**a**) Among various elicitors, YE1 and YE5 elevated resveratrol product levels to >19% and 50% that of the non-elicitation DJ526 seeds (Ctrl). (**b**) YE5 upregulated RS gene by approximately 80% that of the control (DJ526). * *p* < 0.05 compared with the control.

**Figure 2 molecules-27-01951-f002:**
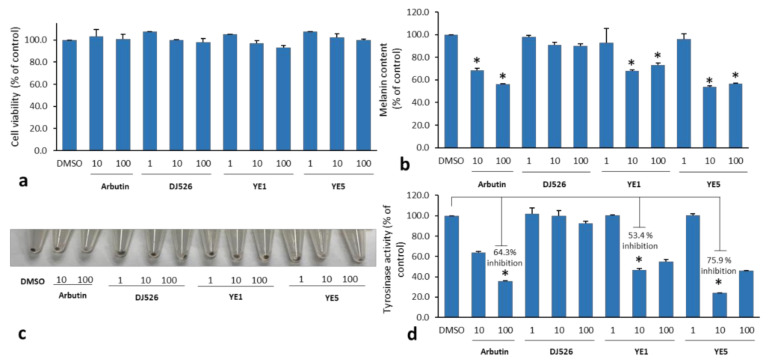
Anti-melanogenesis effects of yeast extract-treated DJ526 seeds extracts. (**a**) Percentage cell viability of melan-a cells after 72 h of exposure to non-elicitor treatment (DJ526) and YE1 and YE5-treated DJ526 seeds extracts compared with the control (DMSO). (**b**,**c**) YE1 and YE5-treated DJ526 seeds showed significantly reduced melanin content and (**d**) inhibited cellular tyrosinase activity in melan-a cells compared with the vehicle control (DMSO) and DJ526 without elicitation, and the inhibitory activities were similar to the positive control (arbutin). Student’s *t*-test was used for statistical analyses. * *p* < 0.05 indicates statistically significant differences compared with the control (DMSO).

**Figure 3 molecules-27-01951-f003:**
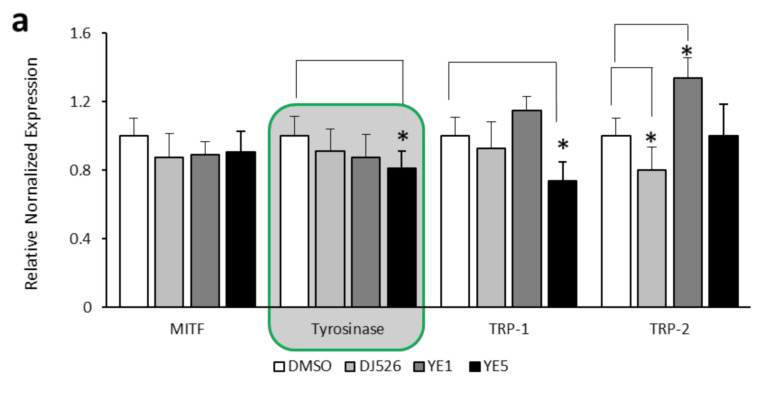
Melanogenesis-related gene and protein expression of crude extracts of DJ526 and 10 mg/mL YE1 and YE5-treated DJ526 seeds. (**a**) mRNA expression of melanogenesis-related genes, including MITF, tyrosinase, TRP-1, and TRP-2, were analyzed by real-time PCR. (**b**) Western blot analysis of protein levels of melanogenesis-related proteins. GAPDH was used as an internal control. Student’s *t*-test was used for statistical analyses, * *p* < 0.05 indicates statistically significant differences compared with the control (DMSO).

**Figure 4 molecules-27-01951-f004:**
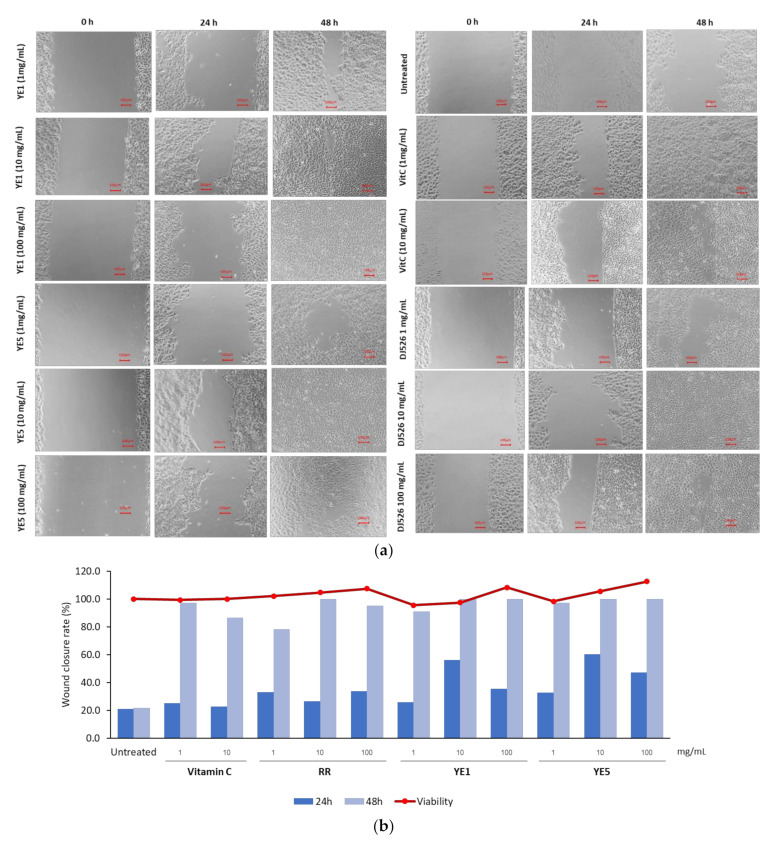
Wound-healing activity of DJ526, YE1, and YE5 treatment. (**a**) Microscopic images of wound gap closure at 0, 24, and 48 h (10× magnification). Scale bars indicate 100 μm. (**b**) Wound closure rate of DJ526, YE1, and YE5 treatments at concentrations of 1, 10, and 100 mg/mL compared with vitamin C (positive control).

**Figure 5 molecules-27-01951-f005:**
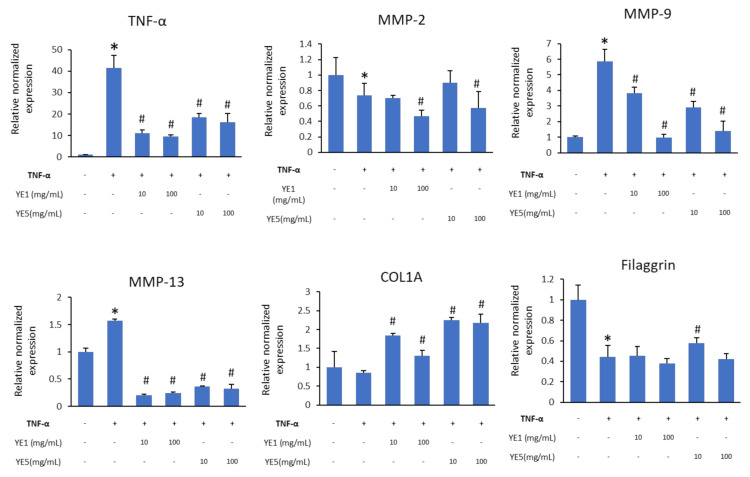
Gene expression of inflammatory mediators, collagen type-1, and skin barrier-related gene in TNF-α-induced HaCaT cells. Student’s *t*-test was used for statistical analyses, * *p* < 0.05 compared with cells not treated with TNF-α. # *p* < 0.05 compared with the control (+ TNF-α).

**Table 1 molecules-27-01951-t001:** Antioxidant activity of DJ526 and elicitor-treated DJ526 seed extracts at various concentrations (1–100 mg/mL).

Elicitor-Treated DJ526 Extracts	Concentration (mg/mL)	Antioxidant Activity (%)	Vitamin C Equivalents of Antioxidant Capacity (mg VEAC/g DW)	IC_50_ (mg/mL)
DJ526	1	3.9 ± 1.39 ^g^	1.04 ± 0.019 ^a^	
	10	28.6 ± 0.11 ^f^	2.66 ± 0.007 ^c^	2.2 ± 0.015
	100	90.7 ± 0.92 ^a^	6.73 ± 0.061 ^i^	
YE1	1	6.3 ± 0.86 ^g^	1.20 ± 0.056 ^a,b^	
	10	36.3 ± 0.76 ^d,e^	3.17 ± 0.05 ^d,e^	2.1 ± 0.006
	100	93.2 ± 0.42 ^a^	6.90 ± 0.028 ^i^	
YE5	1	6.5 ± 0.57 ^g^	1.21 ± 0.037 ^a,b^	
	10	94.1 ± 0.23 ^a^	6.95 ± 0.015 ^i^	1.7 ± 0.008
	100	94.3 ± 0.07 ^a^	6.97 ± 0.005 ^i^	
MJ 0.01 mM	1	7.5 ± 0.13 ^g^	1.2 ± 0.009 ^a,b^	
	10	34.4 ± 1.70 ^e,f^	3.04 ± 0.111 ^d^	3.2 ± 0.010
	100	42.6 ± 0.88 ^c,d^	3.58 ± 0.058 ^f^	
MJ 0.1 mM	1	10.1 ± 0.38 ^g^	1.45 ± 0.025 ^b^	
	10	51.6 ± 4.39 ^b^	4.17 ± 0.288 ^g^	2.5 ± 0.077
	100	56.5 ± 5.35 ^b^	4.49 ± 0.350 ^h^	
JA 0.01 mM	1	7.4 ± 0.25 ^g^	1.27 ± 0.016 ^a,b^	
	10	41.3 ± 7.92 ^c,d,e^	3.49 ± 0.519 ^ef^	3.2 ± 0.402
	100	41.7 ± 5.27 ^c,d,e^	3.52 ± 0.346 ^f^	
JA 0.1 mM	1	7.6 ± 0.25 ^g^	1.29 ± 0.016 ^a,b^	
	10	38.9 ± 4.73 ^b,c^	3.39 ± 0.237 ^e,f^	3.0 ± 0.299
	100	41.3 ± 1.21 ^c,d,e^	3.49 ± 0.079 ^e,f^	

Values are represented as mean ± SD of at least three experiments. Different superscripts for identical parameters indicate significant differences (*p* < 0.05) analyzed using one-way ANOVA with Duncan’s multiple range test.

## Data Availability

All the applicable data have been provided in the manuscript. The authors will provide additional details if necessary.

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
