# Peer review of "Enhanced Anti-Skin-Aging Activity of Yeast Extract-Treated Resveratrol Rice DJ526"

_molecules, 2022, doi:10.3390/molecules27061951_

Round 1

Reviewer 1 Report

Research conducted with resveratrol-enriched rice DJ526 has shown that considering a synergistic effect in the development of GM crops can lead to the generation of unexpectedly beneficial health effects that can be used to prevent and / or treat various age-related diseases.

Research on the elicitation of secondary metabolites and other metabolic processes in higher plants by various elicitors is widely conducted in many research centers around the world. Some secondary metabolites are an important source of bioactive substances in the pharmaceutical industry, they are used in the food industry as food additives, fragrances and many other industries.

The effect of elicitors of resveratrol production in five-day-old DJ526 seeds was investigated in this study. This step has previously been shown to be associated with high resveratrol production.

The potential therapeutic use of DJ526 rice has been well described; however, the use of elication to induce resveratrol production and bioactivity for cosmetic purposes is not well understood. Therefore, the authors used several elicitors to improve the production of resveratrol in DJ526 seeds through elicitation.

The antioxidant activity of the DJ526 seed extracts treated with the elicitor was based on the ability of the extracts to remove radicals that discolored the ABTS ● + radical solution.

A question for the authors: why only this radical was chosen and not, for example, DPPH?

In addition, the treated seeds of DJ526, YE1 and YE5 were selected for in vitro cellular experiments. Antimelanogenesis was tested in melan-A cells. The skin whitening effect of YE was assessed by measuring the melanin content and tyrosinase activity in melan cells. The HaCaT human keratinocyte lineage was used as a cell-based in vitro wound healing model to test the viability of cells exposed to DJ526, YE1, YE5 and vitamin C (positive control) using the Ez-cytox protocol.

Question to authors, please provide details of Ez-cytox protocol.

The authors also showed that YE-treated DJ526 seeds, which produce large amounts of resveratrol, have the potential to promote collagen production in human keratinocytes. Resveratrol and piceid (or polydatin) can promote anti-aging skin by activating sirtuin, which is a key factor in delaying cell aging.

This work provides vital information on the benefits of using GM plants in this case of rice for therapeutic purposes, especially in the development of dermocosmetics and cosmeceuticals. It may be released after some minor revision.

Author Response

A question for the authors: why only this radical was chosen and not, for example, DPPH?

Answer: ABTS was chosen due to the high efficiency detection in plant-containing resveratrol which have been report from the previous study (as shown in discussion, line 258-263, page 9), also ABTS assay was significantly detect in fruits, vegetables compared to those DPPH. Moreover, pigmented solution such as plant extracts (in this study from rice) have been reported that better reflected by ABTS assay than DPPH assay (Floegel et al.,2011; https://doi.org/10.1016/j.jfca.2011.01.008.

Question to authors, please provide details of Ez-cytox protocol.

Answer: In this study we used Ez-cytox kit for testing cell availability. The detail of modified Ez-cytox protocol by referring the product description. (Also provide in materials and methods section through the menuscribe)

  1. Cells were cultured in 96-well plate in a final volume of 100 μl/well culture medium.
  2. Incubated overnight under condition 5% CO2 at 37°C
  3. Mix culture media and testing extracts (DMSO-vehicle control, and DJ526, YE1, YE5 and vitamin C-experimental treatment) and place in each well. Empty wells were excluded for blank. Incubate under condition 5% CO2 at 37°C for 72 hours.
  4. After the incubation, add Ez-cytox 10 μl into each well.
  5. Incubate for 4 hours in culture condition (5% CO2 at 37°C)
  6. Plate was read at the absorbance 450 nm

* Calculation of viability Viability (%) = EXP. – Blank High control – Blank × 100

EXP. : absorbance of a well with cells, EZ-Cytox solution and test solution.
High control : absorbance of a well with cell and EZ-Cytox solution, without test solution.

Blank : absorbance of a well with medium and EZ-Cytox solution, without cell.

Reviewer 2 Report

On the manuscript: Molecules-1618743 “Enhanced anti-skin-aging activity of yeast extract-treated 2 resveratrol rice DJ526

      The present manuscript evaluated the antiaging properties of resveratrol enriched rice DJ526, after elicitation using yeast extract (YE), methyl jasmonate, and jasmonic acid. The results demonstrated that DJ526 seeds elicited using 1 g/L (YE1) and 5 g/L 13 yeast extract (YE5) showed enhanced resveratrol production and the antioxidant activity. YE5-treated 14 DJ526 decreased the melanin content by 46.1% and 37.0% compared with the negative control and DJ526 (non-elicitation), respectively and improved the wound healing activity. YE1 and YE5 significantly decreased the levels of proinflammatory cytokine, TNF-α, and enhanced collagen synthesis in inflammatory cells. The approached topic is very important for the treatment of skin ageing, the experiments performed are interesting but the paper is not ready for publication in its present form and needs major revision.

Comments and suggestions:

  1. The authors should add more details on the pathogenesis of melasma, the role of oxidative stress, inflammation, melanogenesis in this skin condition. What is the relationship with skin ageing? The two diseases are different.
  2. The information needed for the scientific argument of the study is missing. Why they chose to study melanogenesis and wound healing? There are two different processes put under the same umbrella without explaining why they consider them mechanisms belonging of skin aging. The arguments for pathogenesis of skin ageing are missing. There are two mechanisms completely different put together. Please add more details to argue this association.
  3. Please explain why the authors consider MMP as proinflammatory markers?
  4. The discussions have to contain comparisons with other studies from literature.

Author Response

  1. The authors should add more details on the pathogenesis of melasma, the role of oxidative stress, inflammation, melanogenesis in this skin condition. What is the relationship with skin ageing? The two diseases are different.

Answer: I have added more details on the introduction (page2, from line 51). Overproduction of melanin such as melasma, dark spot and the appearance of the mottled are the cause of premature aging. The production of melanin also associated with inflammation and is commonly seen in skin prone to inflammation. To study the effects on skin aging, we should understand both of melanogenesis and skin inflammation.

  1. The information needed for the scientific argument of the study is missing. Why they chose to study melanogenesis and wound healing? There are two different processes put under the same umbrella without explaining why they consider them mechanisms belonging of skin aging. The arguments for pathogenesis of skin ageing are missing. There are two mechanisms completely different put together. Please add more details to argue this association.

Answer: As the objective of this study, we aimed improved skin-therapeutic opportunities in DJ526 seeds, both the skin whitening property and wound healing in rice would be useful in the development of DJ526 extract for dermatological use and the potency in further development as an ingredient in anti-aging cosmetics due to its high antioxidant activity. Thus, wound healing study was examined to increase an impact on the skin-therapeutic opportunities of the DJ526 rice (page 2, line 60).

  1. Please explain why the authors consider MMP as proinflammatory markers?

Answer: MMP or matrix metalloproteinase are inflammatory mediator. MMP9 is associated with collagen degradation after cells are induced inflammation with TNF-alpha (proinflammatory cytokine). Moreover, MMP-2 and MMP-13 have previously been studied in inflammatory cells, so understanding of inflammatory mediator is important to evaluate anti-inflammatory activity.

  1. The discussions have to contain comparisons with other studies from literature.

Answer: Thank you for your comment. In the discussion section, I have added some necessary content that compare our results with other literature. The comparison of resveratrol in resveratrol rice (DJ526) and other resveratrol source are compared in 2nd paragrapgh. In vitro whitening effect and anti-inflammatory effect also have been discussed. Despite, our material is not commonly found and few study, this study has covered the subjected matter that should be discussed.

Reviewer 3 Report

This is interesting theme. To improve the manuscript quality, some of my suggestion as follow:

  1. Figure 1 b. The information about b in the figure is missing and the y axis is better to make it more clear information
  2. IC50 value of antioxidant activity can make the information about significant different between each treatment 
  3. for antioxidant capacity i think better to write it as ascorbic acid equivalent compared with vitamin C equivalent since a lot of type of vitamin not only C
  4. Figure 4. better to put information about unit of 1, 10, and 100
  5. For HPLC analysis please write detail about the gradient system used

Thank you so much

Author Response

  1. Figure 1 b. The information about b in the figure is missing and the y axis is better to make it more clear information

Answer: Thank you for pointing this out, I have added “b” in the figure1, also the y axis has been explained in the figure description.

  1. IC50value of antioxidant activity can make the information about significant different between each treatment 

Answer: Thank you for your suggestion. I have added IC50 value in the column (Table1), also mention in the results (page 3, line 108).

  1. for antioxidant capacity i think better to write it as ascorbic acid equivalent compared with vitamin C equivalent since a lot of type of vitamin not only C

Answer: Thank you for your suggestion. I have added the word “ascorbic acid” to clarify the word “vitamin C” which known as ascorbic acid in line 97 page3. However, vitamin C equivalent antioxidant capacity or VEAC value is commonly used in antioxidant activity measurement which can found in many previous research. Also, it would be clear for readers to not get confused with the first letter “V” from the abbreviation of VEAC where the letter “V” stands for Vitamin C, described that this study compared the capacity of extracts with vitamin C.

  1. Figure 4. better to put information about unit of 1, 10, and 100

Answer: thank you for posting this out, I have added the unit at the end of x axis (figure4, page 7)

  1. For HPLC analysis please write detail about the gradient system used

Answer:  I have revised by adding the gradient program in material and methods section as follow: The gradient program was as follow: 0-37 min, 10% solvent B; 37-38 min, 30% solvent B; 38-45 min, 100% solvent B; 45-50 min, 10% solvent B. (Page 11, line 323)

Round 2

Reviewer 2 Report

The authors have satisfactory answered to all questions mentioned.